

# Agricultural harvesting emissions of ice nucleating particles

Kaitlyn J. Suski[1*], Tom C. J. Hill[1], Ezra J. T. Levin[1], Anna Miller[2], Paul J. DeMott[1], and Sonia M. Kreidenweis[1]

[1]Department of Atmospheric Science, Colorado State University, Fort Collins, 80523, USA
[2] Reed College, Portland, OR 97202 USA

[*]Now at Pacific Northwest National Laboratory, Richland, 99354, USA

*Correspondence to*: Kaitlyn J. Suski (Kaitlyn.Suski@pnnl.gov)

**Abstract.** Agricultural activities can modify natural ecosystems and change the nature of the aerosols emitted from those landscapes. The harvesting of crops can loft plant fragments and soil dust into the atmosphere that can travel long distances and interact with clouds far from their sources. In this way harvesting may contribute substantially to ice nucleating particle (INP) concentrations, especially in regions where agriculture makes up a large percentage of land use. However, a full characterization of particles emitted during harvesting has not been reported. This study characterizes immersion mode INPs emitted during harvesting of several crops in the High Plains region of the United States. The Colorado State University Continuous Flow Diffusion Chamber (CFDC) and the Ice Spectrometer (IS) were utilized to measure INP concentrations during active harvesting of four crops in Kansas and Wyoming. Large spikes of INPs were observed during harvesting, with concentrations over 200 $L^{-1}$ at -30 °C measured during a wheat harvest. To differentiate between mineral and organic components, a novel heating tube method was employed in real-time upstream of the CFDC to deactivate organic INPs in-situ. The results indicate that harvesting produces a complex mixture of organic, soil dust, and mineral components that varies for different crops. Electron microscopy analysis showed that while mineral components made up a large proportion of INPs, organic components comprised over 40% of measured INPs for certain crops at warm temperatures. Heating and enzyme post-treatment of aerosol samples collected for IS processing indicated that bacteria, heat-labile, and heat-stable organics contributed to wheat harvest-produced INPs. These results indicate that plant material and organic particles are a significant component of harvest INPs and their impacts on ice formation in clouds and precipitation on a regional scale should be explored.

## 1 Introduction

Currently, the accuracy of climate change predictions is limited by large uncertainties associated with quantifying aerosol-cloud interactions (IPCC, IPCC, 2013). One step toward narrowing these uncertainties is identifying and quantifying key sources of aerosol particles that can aid in the formation of ice crystals in clouds, termed ice nucleating particles (INPs). INPs are rare in the atmosphere (DeMott et al., 2010) and their sources are not well characterized or quantified. Some INP sources have been identified including, but not limited to, mineral dust (DeMott et al., 2003), lofted biological particles (Pratt et al.,



2009;Creamean et al., 2013), biomass burning aerosol (McCluskey et al., 2014;Prenni et al., 2012), sea spray (DeMott et al., 2015a;Wilson et al., 2015), decaying leaf litter (Conen et al., 2016;Schnell and Vali, 1976), macromolecules on pollen (Pummer et al., 2012), and certain strains of fungi (Fröhlich-Nowoisky et al., 2015;O'Sullivan et al., 2016;Morris et al., 2013). Additionally, changes in emission rates of INPs have been correlated with rain events (Huffman et al., 2013;Prenni et al., 2013)

and high relative humidity (Wright et al., 2014). Recently, soil dust and its ice nucleation potential have gained attention (Conen et al., 2011;Tobo et al., 2014;O'Sullivan et al., 2014;Hill et al., 2016). Soil dust contains both mineral and organic components and it has been suggested that the organic and biological fractions of soil dust are responsible for a majority of its ice nucleation ability (Tobo et al., 2014;Conen et al., 2011;O'Sullivan et al., 2014;Hill et al., 2016). A variety of organic sources of INPs in soils including bacteria, fungi, and other soil organic matter classes have been identified (Hill et al., 2016); however,

a thorough understanding and quantification of INPs from soil organic matter is lacking.

Aside from soils, plant fragments pulverized during harvesting, and biological particles released from the surface of plants can serve as INPs. For instance, large amounts of ice nucleating bacteria have been measured on leaf surfaces (Hill et al., 2014;Georgakopoulos and Sands, 1992), and elevated levels of INPs were observed during active harvesting of a corn field,

some of which were identified as ice nucleating bacteria (Garcia et al., 2012). Harvesting can loft these biological particles into the air (Lighthart, 1984) and once lofted they can travel long distances (Aylor, 1986;Nagarajan and Singh, 1990). Thus, harvesting can be a large daily and seasonal emission source of biological particles and INPs that could have regional impacts on precipitation.

Arable land makes up roughly 11 percent of land surfaces on Earth (FAO, 2010) and in the central United States the majority of land is used for agriculture. Characterizing the sources of INPs in such intensively disturbed land is, thus, essential to accurately predicting their role in cloud development and precipitation events over agricultural regions. Data presented here were collected during harvesting of four crops in the U.S. High Plains over two years at agricultural research centers in Kansas and Wyoming. This work builds upon previous studies of harvest emissions by also utilizing pre- and post-treatments of the

samples, and electron microscopy, to investigate the various mineral, organic, and biological components that contribute to the ice nucleating ability of harvest emissions.

## 2 Methods

### 2.1 Harvest sampling description

Measurements were made at the Kansas State University Northwest Research Extension Center in Colby, KS and at the

University of Wyoming Sustainable Agriculture Research and Extension Center (SAREC) in Lingle, WY. Harvests of soybean, sorghum, and wheat were sampled in Kansas and a corn harvest was sampled in Wyoming. The harvests usually lasted 2-4 hours. Details of the sampling locations and dates are listed in Table 1. A mobile laboratory and gasoline-powered generators



were used to sample in the fields during the harvests. Generators were always positioned downwind of the mobile laboratory. Additionally, the mobile laboratory was positioned downwind from the field being harvested and was repositioned if the plume was no longer reaching the laboratory due to a shift in wind direction or position of the combine harvester. Figure S1 shows an example GPS track of a combine harvester during a corn harvest, and the mobile laboratory position in relation to the

combine. Aerosols were sampled through an inlet comprised of a stainless steel rain hat with a ½" OD stainless steel tube attached. From this tube, y-splitters were used to split the aerosol flow among various instruments. A schematic of the various sampling configurations used during the study is provided in Figure S2.

## 2.2 Aerosol instrumentation

Ice nucleating particle concentrations were measured with the CSU Continuous Flow Diffusion Chamber (CFDC) (Rogers et

al., 2001;Eidhammer et al., 2010). Aerosols are introduced into the CFDC chamber, which has two cylindrical walls that are coated with ice at different temperatures. The temperature difference results in a supersaturation gradient that permits calculation of the supersaturation and temperature at the predicted position of the aerosol lamina ring within a particle-free sheath flow between the walls. For the measurements performed here, the CFDC was operated at 5 % supersaturation with respect to water over a temperature range of -15 to -32 °C. As discussed in DeMott *et al.* (2015b), supersaturation uncertainty

ranges from <1.6 to >2.4% over this range of temperatures. While these settings potentially permit deposition, immersion and condensation-freezing modes of ice nucleation to occur, conditions in the supersaturated (or growth) region of the CFDC emphasize aerosols growing into water droplets via condensation, and then droplets that contain INPs freezing into ice crystals. Hence, data collected during operation in this manner is often compared to methods that explicitly examine immersion-freezing nucleation. Downstream of the growth region, droplet evaporation is stimulated (evaporation region) by holding the two iced

walls at the same temperature to create water sub-saturated conditions. This design feature amplifies the size difference between larger ice crystals and smaller aerosol particles. All particles are sized and counted by an optical particle counter (OPC, Climet CI-3100) and particles larger than 3 μm are counted as ice crystals. A 2.4 μm impactor (50 % aerodynamic cut-size diameter) was used upstream of the CFDC inlet to limit the size of particles entering the CFDC, as large particles (> 3 μm) would otherwise interfere with counting the ice crystals that form in the instrument. The uncertainty in INP concentration

is calculated by adding in quadrature the Poisson counting statistics derived standard deviations of the sample and background periods, which are measured by sampling through a particle filter upstream of the CFDC (Schill et al., 2016). A statistical significance test is also performed on the data. If INP concentrations are greater than the error in INP multiplied by 1.64 (INP > (INP$_{Error}$ * 1.64)), which corresponds to the Z statistic at 95% confidence, then the data is considered statistically significant (Schill et al., 2016).

For select sampling periods, particles that formed ice in the CFDC were collected for chemical analysis via impaction onto electron microscopy (EM) grids (SPI Supplies Coated Grids Formvar®/Carbon, 200 Mesh Nickel) with a 2.9 μm single stage inertial impactor (Kreidenweis et al., 1998). Scanning Electron Microscopy (SEM, Quanta FEG MK2) was used to image the



particles, and Energy Dispersive X-ray (EDX, Oxford Instruments X-Max EDS detector) analysis was performed to obtain elemental composition. Analysis was completed at the University of Wyoming Materials and Characterization Laboratory. Characteristic combinations of elements were identified and then used to group the particles into classes. Particles containing at least one mineral dust marker, such as silicon, aluminum, or iron, were labeled as dust particles. These particles typically

contained oxygen and sometimes carbon as well. Particles with oxygen, carbon, and either sulfur or nitrogen were labeled as organic. If particles contained phosphorous, along with organic markers (O, C, N or S), they were labeled as biological (Pratt et al., 2009). Mixtures of these particle types were labeled as both types. For example, if a particle contained silicon, carbon, oxygen, and nitrogen it was labeled as dust-organic.

The CFDC sample flow rate of 1.5 L min$^{-1}$ sets a limit of detection that restricts its useful temperature range for assessing INP number concentrations. This detection limit is also dependent on the background concentration measured. For the 10 min integrated sampling periods typically used and when background counts are low, this detection limit is ~0.2 L$^{-1}$. To help overcome this limitation, an aerosol concentrator (MSP 4240) was used upstream of the CFDC to concentrate aerosol using virtual impaction (Figure S2b) (Romay et al., 2002). This method has been used in previous studies (e.g., *Tobo et al.,* 2013).

Measurements were taken for 10 minutes with and 10 minutes without the concentrator. A concentration factor (CF) was calculated by taking the number concentration of INPs during concentrated periods divided by the INP concentration during temporally-adjacent non-concentrated periods. During harvesting, there were large spikes in concentration due to the passing of a combine harvester. This made CF calculations difficult when the concentrated and non-concentrated INP values were not equally affected by the spikes. Therefore, the CF calculated during a pre-soybean harvest period in Colby, KS (CF = 90 ± 3)

was used as the CF for all of the harvests. CF uncertainty was calculated by propagating the uncertainties in the INP values used to calculate it. Then, the INP number concentrations during periods using the concentrator were corrected by dividing INP number concentrations by the CF.

The Ice Spectrometer (IS) immersion freezing method uses aerosols collected onto filters over periods of 2-4 hours, achieving

800-3500 L sample volumes that can extend the range of INP measurements to warmer temperatures and a detection limit of ~0.001 INPs L$^{-1}$. Although creating difficulties for comparing methods when higher frequency changes in INP concentrations are occurring, these two methods are complementary, offering colder temperatures and higher time resolution with the CFDC and warmer temperatures and lower INP detection limits with the IS. For IS analysis, aerosols were collected onto 47 mm diameter, 0.2 µm pore diameter (sometimes 0.05 µm) polycarbonate Nuclepore filters (Whatman, GE Healthcare Life

Sciences) fitted within open-faced Nalgene sterile filter units (Thermo Fisher Scientific Inc.). During the wheat and corn harvests, a 2.5 µm cyclone (50 % aerodynamic cut-size diameter at 16.7 L min$^{-1}$, URG Corporation) was also used, upstream of a 47 mm diameter inline aluminum filter holder (Pall Corporation) fitted with a 0.2 µm diameter pore Nuclepore membrane. This limited the size of the particles collected to the same size range as the CFDC. Filters and dissembled filter holders were





cleaned before use by immersion in 10% $H_2O_2$ for 30 min followed by three rinses in deionized water (18 MΩ and 0.2 μm pore diameter filtered), and then dried by removal of excess water and placement on foil in a clean air laminar flow cabinet.

For processing in the laboratory, filters were transferred to sterile, 50 mL Falcon polypropylene tubes (Corning Life Sciences),
7-10 mL of 0.02 μm pore diameter filtered water (Anotop syringe filter, Whatman) with 2 mM KCl (to ensure maintenance of activity of K-feldspar, if present) was added, and the tubes were tumbled end-over-end at 1 cycle s$^{-1}$ for 20 min (Roto-Torque, Cole-Palmer) to re-suspend particles. Measurements of immersion freezing were made on this suspension and 20-, 400- and 8000-fold dilutions of it. Thirty two, 50 μL aliquots of each dilution and a negative control (2 mM KCl) were then dispensed into two 96-well PCR trays (μCycler, Life Science Products), which were then transferred to the cold blocks in the IS. The
trays were then slowly cooled by lowering the temperature at a rate of 0.3 °C min$^{-1}$ from 0 to -27 °C, and the numbers of wells frozen were counted at 0.5 or 1 °C intervals. Cumulative numbers of INPs per volume of liquid as a function of temperature were estimated using the formula $-\ln f_u(T)/V$, where $f_u(T)$ is the proportion of droplets not frozen at a given temperature and $V$ is an aliquot volume (Vali, 1971). Values were then converted to concentrations per liter air samples. Uncertainties are given as binomial sampling confidence intervals (95 %) (formula no. 2, (Agresti and Coull, 1998)). For a detailed description of the
IS see Hiranuma *et al.* (2015).

Ambient aerosols were sized at aerodynamic sizes larger than 0.542 μm using an Aerodynamic Particle Sizer (APS, TSI 3321) and counted using a Condensation Particle Counter (CPC, TSI 3010). The Wideband Integrated Bioaerosol Sensor (Droplet Measurement Technologies WIBS-4A) was used to collect information on fluorescent and biological aerosols. The WIBS-4A,
from here onward referred to as the WIBS, gives fluorescence information in three channels: FL1 (fluorescence at 310 – 400 nm, excited at 280 nm), FL2 (fluorescence at 420 – 650 nm, excited at 280 nm), and FL3 (fluorescence at 420 – 650 nm, excited at 370 nm). Particles from 0.8 to 20 μm are sized by light scattering. The data was classified based on fluorescence signatures into four particle classes: any particles measured with the WIBS (total particles), particles that fluoresced in at least one channel (FP), particles that fluoresced in two channels (termed fluorescent biological aerosol particles, FBAP), and
particles that fluoresced strongly in channel FL1 and weakly or not at all in channels FL2 and FL3 (FP3), as described in Wright *et al.* (2014). The WIBS instrument was non-functional during the corn harvest on November 9, 2015; thus, fluorescence data supplied for corn is from a corn harvest on a different day (November 4, 2015), but at the same field in Wyoming.

## 2.3 Pre- and post-treatments

This work utilized two different types of treatments to tease out the various biological and chemical compositional influences on INPs measured in the ambient environment. Upstream of the CFDC, a tube heated to 300 °C was used to deactivate organic components before they entered the CFDC. The heating tube set up, shown in Figure S2c, consists of two tube furnaces (Thermolyne 21100) placed next to each other in series with a 1-inch diameter quartz tube running through the center of the





tube furnaces. By measuring INP concentrations with and without passage through the heating tube, the fraction of organic INPs, which are deactivated by heating, can be measured in-situ. Previously, heat and peroxide have been used to degrade organic INP components in bulk soil samples and aerosol generated from it post treatment (Tobo et al., 2014;Hill et al., 2016). While useful, these methods do not provide precise time resolution on single particles, which is important for episodic events.

Thus, to enable higher time resolution of single ambient particles, the online heating technique was developed and used in this work. For initial optimization, to ensure that the heating tube method was comparable to the previous bulk heating results, the same soil sample (soil from a sugar beet crop collected in Wyoming) used by Tobo *et al.* (2014) in the bulk heating analysis was aerosolized and run with the heating tube setup. The previous study measured aerosolized pre-treated soil particles size-selected at 0.6 µm using a Differential Mobility Analyzer (DMA, TSI 3080) before sampling with the CFDC. In this study,

particles are size-selected at 0.5 µm. However, this difference in size did not greatly change the results. Figure S3 shows the previous results using the bulk heating method (Tobo et al., 2014) plotted with the results using the heating tube. Data from the heating tube at 300 °C agrees well with results from the bulk heating experiment. This comparison demonstrates that, even though particles only pass through the heating tube for 98 seconds with a flow rate of 1.5 L min$^{-1}$, the heating tube technique is as effective at degrading organic components as the bulk heating method, which entailed heating to 300 °C in an oven for 2

h.

Post-treatments were also applied to the IS filter wash water of the wheat harvest sample to selectively deactivate different INP components. To denature labile organic components (e.g. proteins) an aliquot was heated to 95 °C for 20 min, while to decompose and remove all organic INPs an aliquot of the wash water was digested with hydrogen peroxide. The latter used

the same method as detailed used in McCluskey et al. (2018), except we used a more powerful UV source for hydroxyl radical generation. Briefly, this entailed adding 30% $H_2O_2$ (Sigma Aldrich) to the aliquot to achieve a final concentration of 10%, then immersing the suspension in water heated to 95 °C for 20 min while being illuminated with two, 26 W UVB fluorescent bulbs (Exo Terra) to generate hydroxyl radicals. To remove residual $H_2O_2$, catalase (Cat. Number 100429, MP Biomedicals) was added in 20 µL aliquots to the cooled solution, allowing several minutes between each addition, until no further effervescence

occurred. To lyse all bacteria (including known IN species) another aliquot was incubated with lysozyme to digest their cell walls (lysozyme also hydrolyses fungal chitin oligosaccharides but not the chitin polymer itself). An aliquot of the aerosol suspension was amended to contain 4 mg mL$^{-1}$ lysozyme, 10 mM Tris buffer and 5 mM EDTA (both at pH 8), and incubated at 24 °C for 3 h. For a detailed description of this method see Hill *et al.* (2016).

## 3 Results

### 3.1 Harvest INP emissions

Measurements were made during a soybean harvest on October 14, 2014, a sorghum harvest on October 15, 2014, and during a wheat harvest (June 30 and July 1, 2015, referred to as Wheat 1 and Wheat 2, respectively) in Colby, Kansas, and during a





corn harvest in Lingle, WY on November 9, 2015. Figure 1 shows CFDC and IS INP number concentrations measured during the harvests. The corn and wheat IS data were sampled through a 2.5 µm cyclone to limit the size range of particles that were collected. The soybean and sorghum samples contain the full size range of particles present because no cyclone was used to limit the size of the particles collected. The CFDC INP number concentrations are averaged over three to five minute periods

and the IS INP number concentrations represent the average over the whole harvest sampling period (typically 2-4 hours). For a given CFDC operating temperature, there was a broad range of INP number concentrations for a given harvest due to the nature of harvest sampling: the concentrations vary rapidly in time due to the movement of the combine harvester up and down the field, laterally across, and closer and further away, from the mobile laboratory, and stopping and starting of the harvesting. Thus, the difference in time resolution between the CFDC and IS techniques can explain some of the greater spread in CFDC

data. Even so, the IS and CFDC data generally agree well in the overlap region between -15 and -25 °C. INP concentrations ranged from 0.5 to 147 $L^{-1}$ at -30 °C as measured with the CFDC and the IS data showed a maximum INP concentration of 922 $L^{-1}$ at -25.5 °C during the wheat harvest. In general, these concentrations are very high compared to background INP concentrations and global averages (e.g., DeMott et al., 2010). This result was consistent with limited previous harvest sampling (Garcia et al., 2012), but not consistent when comparing crops harvested or even between harvests of the same crop

(i.e., wheat). Further, measurements from a corn harvest in Nebraska described by Garcia et al. (2012) showed INP concentrations from drop freezing analysis between 30 and 80 $L^{-1}$ at -20 °C and an average CFDC INP concentration of 5.9 $L^{-1}$. In this study, average number concentrations of 0.3 $L^{-1}$ and 3.6 $L^{-1}$ were measured with the IS and CFDC, respectively, at this temperature. While, the CFDC results agree quite well between the two studies, the order or two of magnitude difference between the drop freezing and IS measurements, probably due to the use of the cyclone during this study, which would reduce

the number of large particles that are measured and counted as INPs. Additionally, the distance from the combine harvester during sampling or differences in plant and soil properties at the time of harvesting could also contribute. While the soil was dry during the Nebraska harvest, it was wetter during this study due to recent heavy rain, thus limiting the amount of soil dust kicked up during sampling. These results illustrate the complexity of harvest emissions due in part to varying concentrations in time, distance from the source, and soil moisture.

The shapes of all of the IS harvest spectra are similar, with a "hump" at the warm end (-5 to -22°C), which is accentuated in wheat sample 1. This warm temperature hump, which is a frequent feature in terrestrial INP spectra and is commonly observed in precipitation samples, is suggested to be from biological sources (Petters and Wright, 2015). Interestingly, the crop dust emissions from the wheat field were considered particularly strong due its infestation with rust (Farmer, 2015, Personal

Communication), a parasitic fungal infection. Rust breaks down plant cell walls, which can result in more and finer plant dust particles being produced during harvesting. Furthermore, rust damage to leaf tissues would have allowed many adventitious phylloplane bacteria and fungi to have flourished. IN bacteria have been measured on wheat at populations of $10^8$ $g^{-1}$ of fresh green leaf in Wyoming (Hill et al., 2014) and at 3.5 x $10^6$ $g^{-1}$ of fresh dry leaf at harvest in Colorado (Garcia et al., 2012), and rust has been shown to be IN active at warm temperatures (Morris et al., 2013). Thus, these various biological particles could



be contributing to the INP concentrations seen in the pronounced hump in the IS spectrum for this case. Also, total aerosol numbers, the concentration of fluorescent particles (see below), and INP concentrations on this day were the highest observed in all of the harvest measurements. This suggests that the direct and indirect consequences of the fungal infection of the wheat crop could be contributing to the large number of particles and could be altering the characteristics of the emitted particles.

It should be noted that the CFDC and IS INP spectra have different slopes and the concentrations can be quite different at colder temperatures. The CFDC INP concentrations are generally higher or similar to the IS at warmer temperatures, but lower at temperatures below -25 °C. The reasons for this are not fully understood; however, there are some possible explanations, as discussed in DeMott et al. (2017), and revisited here. Particle conglomerates could break up while in the IS wash water, which could provide more INPs in the bulk solution than are present as single particles measured in the CFDC. Alternately, small ice nucleating entities (INEs), such as protein complexes (Hartmann et al., 2013) or macromolecules on pollen (Augustin et al., 2013;Pummer et al., 2012), could be present on and released from the particle surfaces and these INEs might be especially active at lower temperatures. Size could also play a role, as INP sizes are generally larger during harvests (Mason et al., 2016) and are not as effectively sampled into the CFDC. During the soybean and sorghum harvests no size restriction was placed on the IS filters, thus larger particles (> 2.5 µm) are underrepresented in the CFDC data as compared to the IS. This is less of a concern, and indeed may be reflected in the data, for the corn and wheat harvests because a 2.5 µm cyclone was used to restrict particle sizes on the IS filters. Finally, there may be a time dependence of ice nucleation that accentuates differences between the CFDC and IS measurements at lower temperatures. Particles are in the CFDC growth region for approximately five seconds, while they are at a particular temperature for several minutes in the IS. However, previous results suggest there is little temperature dependence to stochasticity (Wright and Petters, 2013), and thus it is unlikely that the time difference is the cause of the discrepancy that occurred at the lowest temperatures.

### 3.2 Fluorescent particles

Fluorescent and biological particle concentrations and types measured by the WIBS were grouped into particle classes as described in the methods section. These data are shown in Figure 2 and Table 2. Interestingly, FBAP were observed before the start of the soybean harvest period, which has been termed the "pre-soybean harvest" period. During the soybean harvest, the FBAP percentage increased from 6.8 % during the pre-soybean harvest period to 17.8 %, indicating that additional biological particles were emitted during the harvest. This pre-harvest period was likely strongly influenced by harvesting in the region even though we were not directly in a fresh harvest plume. The corn harvest produced the largest percentage and concentration of FBAP particles out of all of the harvests sampled (33.5 %, $107.7$ L$^{-1}$). These results suggest an abundance of biological particles are emitted during corn harvests, which is supported by a previous study that showed corn harvests emit bacteria (Garcia et al., 2012) and could also indicate that more plant fragments are emitted during corn harvests than for other plant harvests. While wheat harvests emitted the highest number of fluorescent particles, they had the lowest percentages of FBAP and Wheat 2 had the lowest concentration of FBAP (2 L$^{-1}$). This indicates that the wheat emissions only fluoresced in one





channel and could point to the greater presence of plant material or soil dust, as opposed to other biological particles. Lignin is present in wheat and absorbs at 280 nm and emits at ~360 nm, which would give a signal at FL1 (Albinsson et al., 1999). Wheat lignin also autofluoresces with excitation at 330-385 nm and detection at 420 nm, so it will give a signal at FL3. Therefore, if wheat lignin made up a bulk of the emitted particles they might show up as FBAP; however, FBAP made up a

low percentage of particles. This suggests that the bulk of the fluorescent wheat particles lack lignin and could be non-lignified plant cells or dead microbes.

The FP3 particles did not make up a significant percentage of particles, except during the first wheat harvest and the corn harvest, and average concentrations ranged from 5 L$^{-1}$ in the pre-soybean harvest period to 193 L$^{-1}$ during the Wheat 1 harvest.

These particles have not been biologically or chemically identified, but have been shown to correlate with INP concentrations (Wright et al., 2014). FP3 is indicative of the presence of tryptophan and the absence of NADH and could indicate dead plant and microbial material containing protein (e.g., dead phloem cells, dead bacteria, and fungi). Figure 2 and Table 2 also include IS INP concentrations at three temperatures for comparison to the fluorescent particle concentrations. In general, the INP concentrations measured without a cyclone at -20 °C are on the same order of magnitude as the FP3 concentrations, and thus

FP3 concentrations show potential as an indicator of INP concentration at -20°C. The wheat 1 case is somewhat of an outlier with the highest INP concentration at -15 °C, which was on the same order of magnitude as the FP3 concentration. Overall the best agreement was between the FP concentrations and the -25 °C INP concentrations, which agreed very well for the 3 samples that had IS data at -25 °C.

**3.3 Chemical composition of INPs**

To chemically characterize the harvest INPs, particles were collected via impaction onto SEM grids downstream of the CFDC. During the sorghum harvest, mineral dust made up 41 % of INPs at -17°C, as shown in Figure 3a. Organic material also made up a large fraction, 29 %, of the INPs along with mixtures of dust and organics (13 %) and a small percentage of purely biological particles (2 %). Images of the particles, shown in Figure 3b, include interesting structures that could be indicative of plant material or biological particles. There were plate-like structures with potassium, thread-like filaments containing

silicon, which could be plant material (Lux et al., 2002), in addition to flaky structures with the elemental composition of mineral dust. These results indicate that organic components make up a large percentage of sorghum harvest INPs at -17°C.

SEM-EDX data from the corn harvest at -27°C, shown in Figure 4a, indicates mineral dust again comprised a significant portion of INPs (32 %), along with dust mixtures with organics (Dust-Org, 19 %), biological particles (Dust-Bio, 13 %), and

sulfate (Dust-S, 10 %). Additionally, there was a significant amount of biological particles (18 %), which were identified by the presence of carbon, nitrogen, and phosphorous (Pratt et al., 2009). Many of the measured INPs also had structured forms similar to the sorghum harvest emissions. These particles had oblong, granular shapes and some appeared to have tiny hairs on the surface, suggesting they were of biological origin. The large percentage of biological INPs agrees well with previous



measurements during corn harvests, which also showed the presence of several genera of bacteria among CFDC residuals (Garcia et al., 2012); this included 19 IN bacteria L$^{-1}$ air, quantified directly using quantitative PCR of the *ina* gene. While SEM-EDX elemental compositions indicated the presence of biological particles, a full characterization of the biological components cannot be achieved with this method. Future work will utilize post treatments of the filtered particles to further

identify the types of biological particles that served as INPs. The WIBS data also revealed that the corn harvest produced the largest fraction and concentrations of fluorescent biological aerosol particles (FBAP) out of all the sampling locations. Taken together, these results indicate that organics and biological particles, along with mineral dust, make up a large percentage of harvest INPs between -17 and -27°C. SEM samples were not collected during the soybean and wheat harvests; therefore, a comparison of organic and mineral components cannot be directly assessed from these crops.

**3.4 Inferences regarding INP compositions through use of heat and post-treatments**

In situ heating during real-time CFDC measurements was utilized to assess the contribution of minerals and organics to INPs emitted from harvests. Heating at 300 °C has a similar impact on organics as peroxide digestion and will degrade heat-labile organics and biological particles (Tobo et al., 2014), and thus a comparison of heated and non-heated INP concentrations reveals the percentage of organic versus inorganic INPs. SEM-EDX results presented in Figure 4b show the chemical changes

in INPs that occurred with heating at 300°C during the corn harvest. The percentage of mineral dust increased, which is expected because as organics are degraded with heating minerals will remain IN active and make up a larger percentage of INPs. The percentage of biological INPs was reduced from 18 % to 7 %, but they were not totally deactivated. The heat treatment dramatically reduced the percentage of Dust-Bio INPs from 13 % to 1 %, which suggests the IN-active biological components were degraded with heat, suggesting that the biological components played a larger role in the IN activity than

the minerals within this class. The latter scenario is consistent with previous studies that show organic and protein residues on mineral surfaces can enhance the ice nucleation ability of the minerals (O'Sullivan et al., 2016;O'Sullivan et al., 2014;Conen et al., 2011). Interestingly, the percentage of Dust-Org particles was not reduced with heating. This could indicate that the organics that were internally mixed with minerals were not susceptible to heat at 300 °C, or that after the organics were degraded the INP activity was unchanged because the minerals in these mixed particles were serving as the active sites for ice

nucleation. Alternatively, heating the Dust-Org particles could have evaporated off some volatile organics uncovering active sites on the dust. This study cannot differentiate between those scenarios, but future studies should investigate the physical changes caused by heating, including how heating might change mixing state and surface morphology.

The quantitative changes to ice nucleating ability with heating are shown in Figure 5. The fraction of INPs with respect to the

concentration of total particles larger than 0.5 µm, as measured with the CFDC OPC ($n_{0.5\mu m}$), is plotted on the y-axis. INP fraction is shown instead of INP concentration to allow for direct comparison between heated and non-heated sampling periods. This is because, there were large changes in particle concentrations due to sampling in and out of the harvesting plume and changing wind directions, which complicate a direct concentration comparison. This figure displays only statistically





significant data points, as determined with the significance test described in the Methods section, unless otherwise noted. Results indicate that heating had a large impact on INP number concentration for soybean harvest emissions as cold as -25 °C. Similarly, for the sorghum harvest, INPs were reduced by heating to below detection levels at -18 °C, but a smaller impact was noted at temperatures ≤-22 °C. During the corn harvest, heating reduced the fraction of INPs at warm temperatures (-19

°C) to below the instrumental detection limits, but at colder temperatures (-28 °C) there was only a slight change in the fraction. A similar situation appears for the wheat harvest, although data were not collected for heating trials at below -22 °C. These results suggest a general, albeit variable, impact in which organic (including biological) particles from harvesting exert more influence at warmer temperatures, while at colder temperatures mineral dust components likely dominated the ice nucleation activity. Further characterization of the emissions is necessary to identify the nature of the organic particles, but these results

suggest harvest emissions are distinct for different crops.

The observed decrease in INPs with heating is presented in a different way in Figure 6. Fractional change in INPs is shown for each temperature and crop in cases where heating measurements were made. At temperatures between -17 and -19 °C (the warmest temperature accessible for comparison via CFDC data), all of the harvest samples had large decreases in INP activity

with heating. The fractional changes at these relatively warm temperatures were between -0.7 and -0.98, which suggests that a large percentage of these warm temperature INPs are of organic or biological origin. At -32°C, the INPs fractionally decreased by ~0.5 for all crops indicating that minerals and possibly 300 °C heat-stable organics are contributing up to 50 % of the INPs. However, the fact that there was still a reduction in INP at these cold temperatures agrees with previous results that showed that organics contributed significantly to the INP population of soil dust even at cold temperatures where,

traditionally, minerals are expected to dominate the activity (Tobo et al., 2014).

During the wheat harvest, heat treatment resulted in a 98% reduction in INP number concentrations at -18 °C. This suggests that biogenic particles make up almost all of the INPs at temperatures ≥-18 °C. Additional focus in the wheat sampling was placed on evaluation of the contributions to this degradation observed in situ and in real time with the CFDC. Post-treatments

on the wheat harvest sample via IS immersion freezing measurements, shown in Figure 7, revealed a variety of biological/organic INP compositions contributing to the IN activity >-20 °C, along with an underlying mineral or non-organic contribution to the IN activity, as suggested by the dashed grey line. Lysozyme digestion indicated that bacteria likely contributed foremost to the INP population. By digesting bacterial cell walls, lysozyme will cause rupturing of all bacteria. For the known species of ice nucleation active bacteria (eg, *Pseudomonas syringae*, *Pantoea agglomerans* and *Xanthomonas*

*campestris*) clusters of the protein anchored in the outer membrane will, as the outer membrane disintegrates, disaggregate into smaller clusters active at ~-7 to -10 °C or into single proteins active at -12 to -13 °C (Govindarajan and Lindow, 1988). However, in the wheat harvest sample the effect was observed to as cold as -21 °C, suggesting that other, as-yet unidentified, IN bacteria were not only present but abundant in the wheat dust. While WIBS data suggested that <1% of particles were FBAP, the lysozyme digestion shows that a large amount of bacteria was generated from the harvest.





Bulk heating of the IS sample to 95 °C resulted in a larger reduction of INPs that can be attributed to heat-labile INPs, such as proteins in bacteria and fungi, on the plants and in soil dusts raised by the harvester. There was also a modest amount of organic material that was not susceptible to 95 °C heat, but was degraded with peroxide digestion, that was contributing to INP

concentrations, and is evident in the shaded regions between the red markers and the black markers in Figure 7. If arable soil dust contributed largely to the INP concentrations, peroxide treatment would show a greater reduction in INP than was observed here. The large reduction due to heating indicates that biogenic particles make up a large percentage of INPs at temperatures warmer than -18 °C. These biogenic particles come from a variety of sources, which highlights the complex nature of INPs emitted from agricultural and soil perturbation activities. No one particle type can accurately describe the nature

of INPs for agricultural areas in general, but rather a mixture of biogenic particle types best represents these emissions. The findings in this study suggest that harvesting and plant litter emissions stimulated by wind at the surface, provide the most viable explanation of the ubiquity of heat-labile INPs in the High Plains boundary layer even in the absence of harvesting, as found by Garcia et al. (2012).

## 3.5 Discussion and atmospheric implications

Results presented herein, especially those shown in Figures 6 and 7, emphasize the potential need to include harvesting INP emission impacts in regional cloud models to assess their subsequent impacts on clouds and precipitation in both agricultural and naturally-vegetated regions. Harvesting emits mineral, organic, and biological particles into the atmosphere in large quantities. $PM_{10}$ emission factors ranging from 10 to over 1000 kg km$^{-2}$ have been reported for different crops harvested in California, and these emission factors vary based on crop, relative humidity, and soil moisture (Flocchini et al., 2001). A full

characterization of the emitted organic matter is beyond the scope of this work and would involve intensive chemical, biological, and plant pathological investigations. Even from a single source such as harvesting, there are several distinct inputs including but not limited to pulverized plant tissues, dust, bacteria, fungi and other biological particles present on plant surfaces, various biological, organic and mineral INPs lofted from the soil, and even residual fertilizer on the soil surface. This complex combination of sources is difficult to untangle, especially because it can change with geographic location, crop type,

plant and soil states, environmental conditions during harvesting, and year-to-year differences in the many parameters. Additionally, different pathogens can grow on the crops, as was shown with rust-infected the wheat crop sampled in Colby, KS. All of these factors can change the ratio of mineral to organic components in the INPs, which has implications for how these emissions should be represented in models.

To assess the ability of existing INP parameterizations to model harvesting INP concentrations, the measured CFDC and IS INP number concentrations were compared to predicted INPs using parameterizations for average global INP concentrations (D10) (DeMott et al., 2010), mineral dust (D15) (DeMott et al., 2015b), and biological particles (T13) (Tobo et al., 2013) (Figure 8). The D10 and D15 parameterizations predict INP concentrations at a given temperature based on particle number





concentrations above 0.5 µm ($n_{0.5µm}$). For comparison to the IS, particle number ($n_{0.5µm}$ from the CFDC) was averaged over the IS sampling times. The D15 parameterization results presented here do not include the factor of 3 increase suggested in DeMott *et al.* (2015b) for use in predicting atmospheric concentrations of relevance to clouds because comparisons here are made to uncorrected CFDC and IS INP concentrations. The T13 parameterization uses biological particle concentrations,

derived from the WIBS FBAP concentrations in this study, instead of $n_{0.5µm}$, to predict INP concentrations. FBAP concentrations were averaged over the CFDC and IS sampling times to compare to CFDC- and IS-derived INP concentrations, respectively. In applying WIBS FBAP within the Tobo *et al.* (2013) parameterization, we must note that FBAP concentrations used to develop the parameterization were based on an ultraviolet aerodynamic particle sizer (UV-APS) that senses FBAP at sizes above 0.5 µm, while the WIBS FBAP signal is for >0.8 µm particles. Hence, we expect that predicted values may be

somewhat underestimated in this case. Note that all temperatures are integrated into such an analysis, so that biases may enter due to changes in the contributions of different compositions at different temperatures, as has been discussed. Also, the CFDC data presented here cover a narrower temperature range (-17 to -32 °C) than that used in developing these parameterizations (e.g. -9 to -34 °C for D10).

Comparisons shown in Figure 8 indicate that different crops have different relationships with CFDC-derived $n_{0.5µm}$ as described by different parameterizations. For instance, the corn CFDC and IS with cyclone INP data is predicted most accurately by the D15 parameterization. D15 is used to model dust INP activity, thus the correlation suggests that dust was serving as a source of INPs during the corn harvest. The SEM-EDX results presented in Figure 4a confirm this and show that dust and dust mixtures made up 74 % of INPs measured at -27 °C. However, the IS data for all sizes (no cyclone) is predicted best with the

D10 parameterization. The WIBS instrument was inoperable during the corn harvest; therefore, the T13 parameterization could not be tested against the corn data.

The first wheat harvest (Wheat 1) INP concentrations are predicted well with the D10 parameterization for CFDC and IS with cyclone data. This "global" INP parameterization represented a diverse range of INP sources (i.e., not distinct to one source), which may explain why it captures the diverse range of INPs that wheat harvests emit. Furthermore, this again supports the

idea that elevated INP activity at temperatures higher than -20 °C observed in data compilations like DeMott et al. (2010) have their major sources from plant and microbial INPs. The cyclone-IS INP concentrations are predicted best by the D15 parameterization, although they are consistently over-predicted. The relationship between $n_{0.5µm}$ and CFDC INP during the second wheat harvest, Wheat 2, is not captured by any of the tested parameterizations. This could indicate changing emissions

throughout the harvest or a mixture of minerals, organics, and biological particles that does not have a consistent relationship between $n_{0.5µm}$ or FBAP and INP. It might also suggest a non-fluorescing population or an especially active biological population not being represented by the T13 parameterization, which was modeled on data collected in a region rich in fungal spores and which might not capture the behavior of other biological types. To explore this scenario, the markers in Figure 8 were colored by CFDC and IS operating temperature and are displayed in Figure S4. The Wheat 2 data points, as well as some





Wheat 1 and soybean points, that are not well predicted by T13 are mostly at warm temperatures (around -20 °C). Thus, this could indicate a non-spore biological particle type was contributing to the INP population during these harvests.

The soybean harvest was modeled well by both the D10 and D15 parameterizations for both CFDC and non-cyclone IS data; however, the T13 parameterization also accurately predicted the CFDC data except for a few points. This might suggest that the soybean emissions had contributions to INPs from both dust and biological particles. The soybean emissions had a large percentage of FBAP (17.8 %) according to the WIBS and a strong reduction in INP activity after heat treatment, which is indicative of biological INPs. However, the heat did not totally wipe out the IN activity, which suggests the presence of minerals in the INPs as well.

Sorghum CFDC INP concentrations are modeled well by both the D10 and T13 parameterizations, while the IS non-cyclone data was best represented by D15. However, there is a hump in the data between -15 and -18 °C, which is better predicted by T13. This again suggests there is a mix of particle types including mineral dust and biological INPs present. Indeed ~ 11.8 % of particles measured on the WIBS were FBAP, and heating reduced the INP activity of the sorghum to below detection limit at -17 °C and by 63% at – 32 °C, indicating a strong organic contribution to INPs. SEM-EDX analysis of INPs active at -17 °C show 41% of INP were mineral dust and, while a small fraction (2%) of particles were of biological origin, a large percentage of INPs had organic components (42 %). Some of these organic particles could be from biological sources, but based on their low (undetectable) levels of phosphorus were labeled as organic. Phosphorus is often a limiting nutrient in plants and is re-mobilized into living tissues or seeds when plants are senescing. The high organic but low biological signature could indicate that phosphorous was relocated from leaf and stem tissues to the sorghum grain before the harvest, which has been observed in sorghum when phosphorous is limited (Roy and Wright, 1974). Images of some of these particles (Figure 3B) show structured shapes indicative of biological origin. This evidence points to the importance of organic and biological particles as well as mineral dust serving as INPs during the sorghum harvest.

The results presented here suggest that different crops have different relationships between aerosol number concentrations and INP concentrations. No single INP parameterization accurately predicts INPs released during harvest periods for all crops, but both D10 and D15 could be used in agricultural regions to predict ambient INP concentrations during harvest months, given measurements and/or forecasts of aerosol concentrations. FBAP concentration data is not readily available, and thus the comparison to the T13 parameterization is provisional at this point.

The large seasonal increases in harvest emissions could have effects on precipitation, especially in the Plains states where deep convection is frequently occurring. Several modeling studies have investigated the effects of increased aerosol concentrations on convection. One study showed that increases in aerosols modify storm structure but have minimal effects on warm-front precipitation (Igel et al., 2013), while another suggested deep convection in the Great Plains is modified by larger aerosol




loading, by raising cloud-top height in mixed-phase clouds and increasing precipitation rates in clouds with large amounts of liquid water (Li et al., 2011). Increases in biomass burning aerosols have been linked to increases in severe weather (Wang et al., 2009) and the likelihood of tornado formation (Saide et al., 2015), while mineral dust has been shown to have competing effects on squall lines with an overall weakening due to larger dust concentrations (Seigel et al., 2013). It is important to note

that these studies have focused on the effects of aerosols serving as cloud condensation nuclei and have not included aerosols serving as INPs. The varying effects of aerosols on convection highlights the need to further investigate these scenarios and include INPs into these simulations which could change the results and lead to a better representation of clouds and precipitation in agricultural regions in models.

## 4 Conclusions

Measurements made during the harvesting of four crops in the Great Plains indicate that highly complex mixtures of different organic particle types along with mineral components make up the spectrum of activity in harvest-derived INPs. SEM-EDX analysis confirms the presence of organic components in the harvest INP emissions as well as biological particles, mineral dust, and mixtures of these types. High heat tests suggested contributions of both labile and stable organic INPs over the full temperature range measured, accounting for up to half of the INP activity even at -30 °C, but dominating at temperatures above

-20 °C for all crops. Soybean harvest emissions showed the largest contribution of organic components at colder temperatures (-32 °C), while corn harvests produced the largest fraction of biological particles in the total aerosol and showed a large fraction of biological INPs even at –27 °C.

Organic particles, especially those of biogenic origin, contribute substantially to the ice nucleating efficiency of harvest

emissions. This was demonstrated by the effect of heating, which greatly reduced INP concentrations for all crops, with the most pronounced effects at warm temperatures. For example, during the wheat harvest, CFDC INP concentrations at -18 C were reduced by 98% with heat treatment. Post-treatments on the wheat harvest sample indicated the presence of IN active bacteria, mineral dust, and an extraordinarily high proportion of heat (95 °C) labile (e.g., proteinaceous) INPs. The large contribution of heat labile material to INPs is unique for the harvest emissions and has not been observed as being so abundant

in soil dusts (Hill et al., 2016). A small amount of 95 °C heat stable organic INPs that were degraded only with peroxide digestion were also observed. Heat stable organics make up a larger fraction of arable soil dust than were observed here, again suggesting that harvest emissions include plant fragments and other biogenic particles not commonly found in soil dust.

With the ultimate goal of incorporating these data into cloud models, INP parameterizations were used to compare predicted

and measured INP concentrations. These comparisons suggested that INP emissions from several crops are complex mixtures of various types of organic, mineral, and biological particles. The inability of the T13 parameterization to predict warm temperature INPs for several crops is due to the low amount of FBAP observed and suggests the presence of unidentified warm





temperature INPs that are distinct from the spore-dominated scenario in Tobo et al. (2013). Due to the variety of components that contribute to the INPs, the complexity of the INP spectrum is not accurately modeled by existing INP parameterizations. However, the D10 and D15 parameterizations could be used to give estimates of INP in agricultural regions. WIBS data can also be used to give estimates of -20 and -25 °C INP concentrations using FP3 and FP concentrations, respectively. Corn,

5    soybean and wheat are the top three most planted crops in the United States. Over 2014 and 2015, corn, soybean, wheat, and sorghum crops were planted over 960,000 sq km of land in the United States alone (National Agricultural Statistics Service). Harvest emissions can have a large impact on clouds in agricultural regions and this characterization of harvest-emitted INPs can be used to inform quantitative models using aerosol concentration inputs and will hopefully lead to a better understanding of the role of harvest-emitted INPs in convective clouds in these regions.

10   *Data availability.* The data used in this manuscript are available in a digital library at CSU (https://dspace.library.colostate.edu) under identifier: https://hdl.handle.net/10217/187173.

*Competing interests.* The authors have no conflict of interest.

*Acknowledgments.* Funding for this work was provided by NSF grant AGS1358495. Anna Miller was funded by the Reed College Opportunity Fellowship. Special thanks to Norbert Swoboda-Colberg for SEM-EDX analysis, and to Larry Howe, Bob Baumgartner, and Kelly Wiseman at SAREC, and Freddie Lamm, Dan Foster, and Marv Farmer at the KSU NW Research Center for their help with coordinating the harvest measurements.





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

| Sample | Location | Latitude, Longitude | Elevation (m) | Sampling Date |
|---|---|---|---|---|
| Pre-Soybean Harvest | Colby, KS | 39.394, -101.066 | 966 | 10/14/14 |
| Soybean | Colby, KS | 39.394, -101.066 | 966 | 10/14/14 |
| Sorghum | Colby, KS | 39.394, -101.066 | 966 | 10/15/14 |
| Wheat 1 | Colby, KS | 39.394, -101.066 | 966 | 6/30/15 |
| Wheat 2 | Colby, KS | 39.394, -101.066 | 966 | 7/1/15 |
| Corn | Lingle, WY | 42.126, -104.403 | 1309 | 11/9/15 |

**Table 1. Sampling locations, elevations, and dates for all of the harvests are given.**



| Sample | FP % | FP3 % | FBAP % | FP (L⁻¹) | FP3 (L⁻¹) | FBAP (L⁻¹) | INP -15 °C (L⁻¹) | INP -20 °C (L⁻¹) | INP -25 °C (L⁻¹) |
|---|---|---|---|---|---|---|---|---|---|
| Pre-Soybean Harvest | 85.7 | 3.6 | 6.8 | 128.6 | 5.4 | 10.2 | 0.08 | 1.6 | *62 |
| Soybean | 81.2 | 4.9 | 17.8 | 156.9 | 9.5 | 34.4 | 0.26 | 3.0 | 180 |
| Sorghum | 88.5 | 2.6 | 11.8 | 348.5 | 10.1 | 46.6 | 0.5 | 3.5 | 180 |
| Wheat 1 | 90.5 | 11.1 | 0.7 | 1580.0 | 193.2 | 12.7 | 76/**3.1** | 180/**4.7** | **610** |
| Wheat 2 | 65.7 | 4.5 | 0.3 | 415.1 | 28.7 | 2.2 | **0.02** | **1.2** | **91** |
| Corn | 88.8 | 17.8 | 33.5 | 285.3 | 57.2 | 107.7 | 2.7/**0.16** | 8.1/**0.33** | **29** |

**Table 2. WIBS data collected during the harvests showing the percentages and concentrations of fluorescent particles (FP), particles that fluoresce in channel 1 (FP3), and fluorescent biological aerosol particles that fluoresce in two channels (FBAP). INP concentrations measured with the IS with (bold) and without (normal) a cyclone are presented at -15, -20, and -25 °C. The pre-soybean harvest INP concentration is given at -24.5 °C and is indicated by the asterisk.**





**Figure 1. INP number concentrations divided by the concentration factor (CF) measured using the CFDC (squares) and IS (circles) during four harvests are shown including soybean (a), sorghum (b), wheat (c), and corn (d). The smaller squares represent particles sampled on the concentrator, while the larger squares are sampled without the concentrator. Both significant and non-significant data are shown. (c) and (d) are data collected through a 2.5 µm cyclone.**



**Figure 2. WIBS data showing the concentration (circle markers) and percentage (bars) of three different classes of fluorescent particles (FP, FP3, and FBAP) during the each harvest and one pre-soybean harvest period. The corn WIBS data was collected from the same field but on a different day than the corn IS data. The IS INP concentrations for –15 °C (light grey), -20 °C (dark grey), and -25 °C (black) are shown in diamond markers for non-cyclone samples only.**


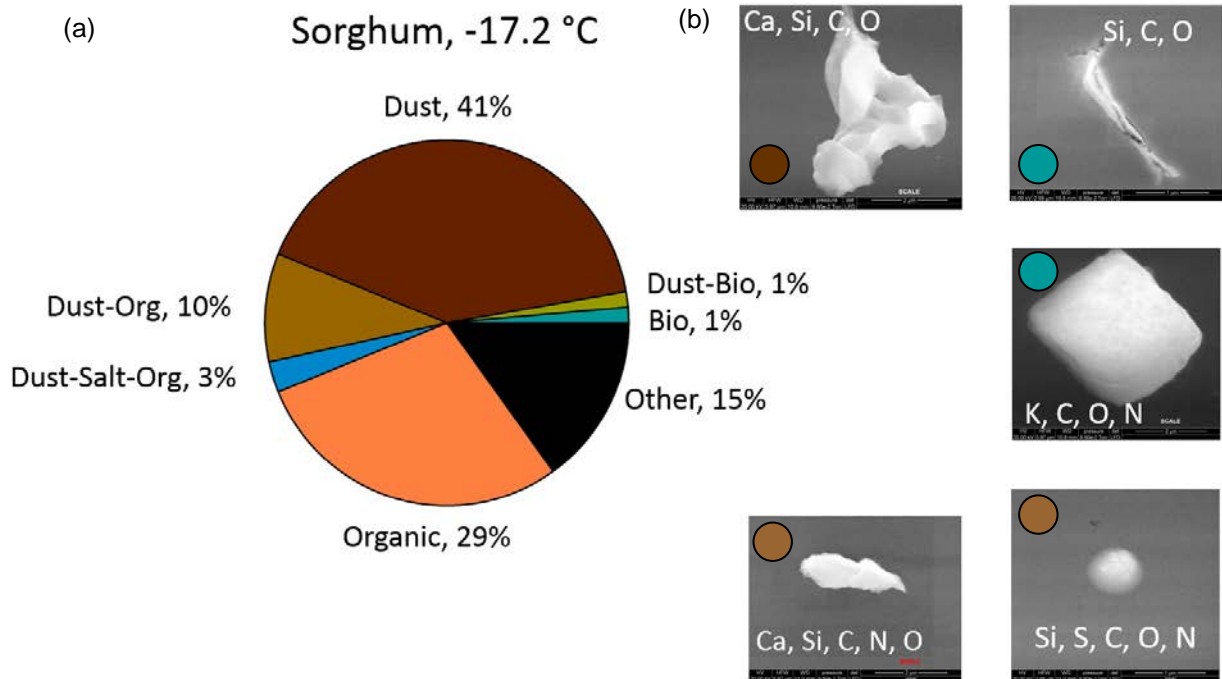

**Figure 3.** Relative amounts of different particle types collected via impaction and analyzed using SEM-EDX (a) and example SEM images with the corresponding elemental composition measured with EDX shown in white (b). The colored circle in the left corner of the images indicate which chemical class the particles were classified as. These data were collected during a sorghum harvest with a CFDC operating temperature of -17 °C.




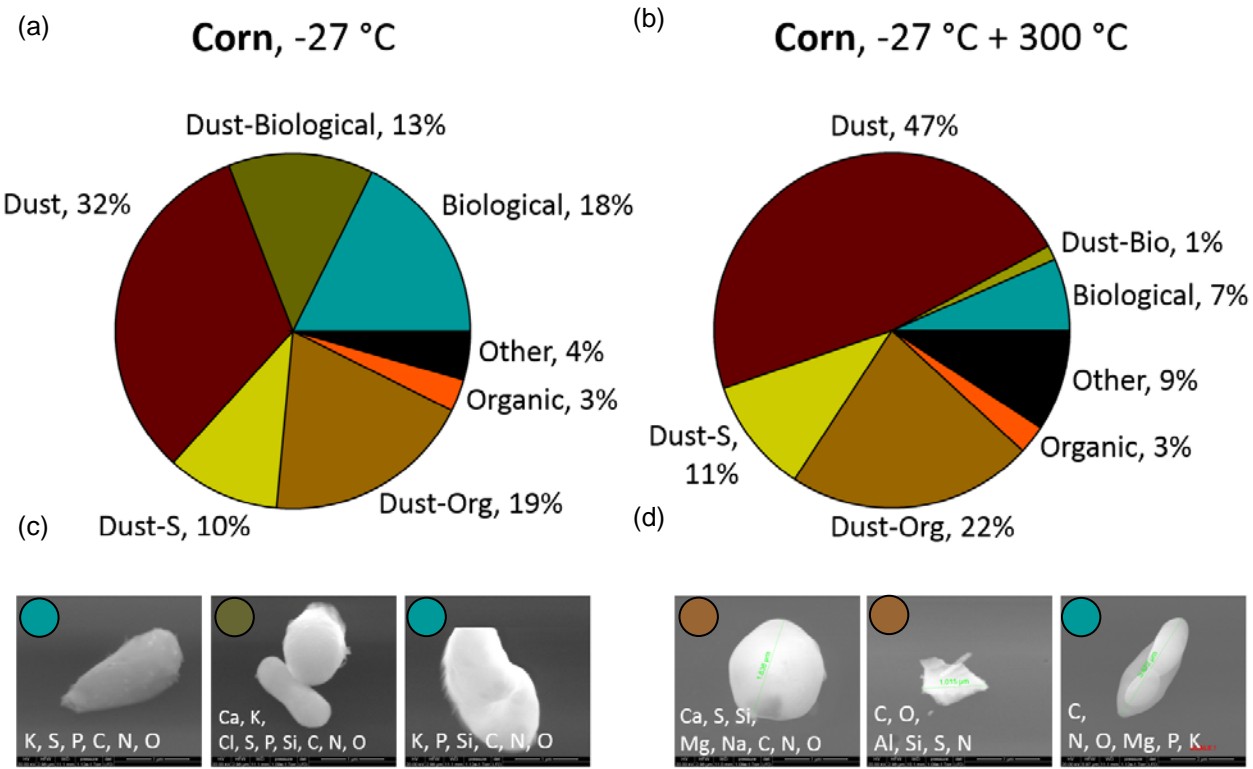

**Figure 4. SEM-EDX data collected during a corn harvest. (a) and (b) show relative amounts of each particle type as sampled using EDX, while (c) and (d) show example images of particles analyzed with SEM. The colored circle in the top left corner of the images indicate which chemical class the particles were classified as. Elemental composition is given in white on each of the example images. The data were collected at a CFDC operating temperature of -27 °C with the left hand side (a, c) showing data without heat and the right hand side (b, d) showing data for particles that had passed through a heating tube at 300 °C upstream of the CFDC.**





**Figure 5. The fraction of INPs out of the total number of particles greater than 0.5 μm as measured by the CFDC ($n_{0.5\mu m}$) is plotted against CFDC operating temperature for four crop harvests: soybean (a), sorghum (b), wheat (c), and corn (d). Data collected through the heating tube at 300 °C is shown in red and non-heated data is in blue. The larger markers represent periods sampled without the concentrator and the smaller markers represent periods sampled through the concentrator. Note the difference in scale on the y-axes.**





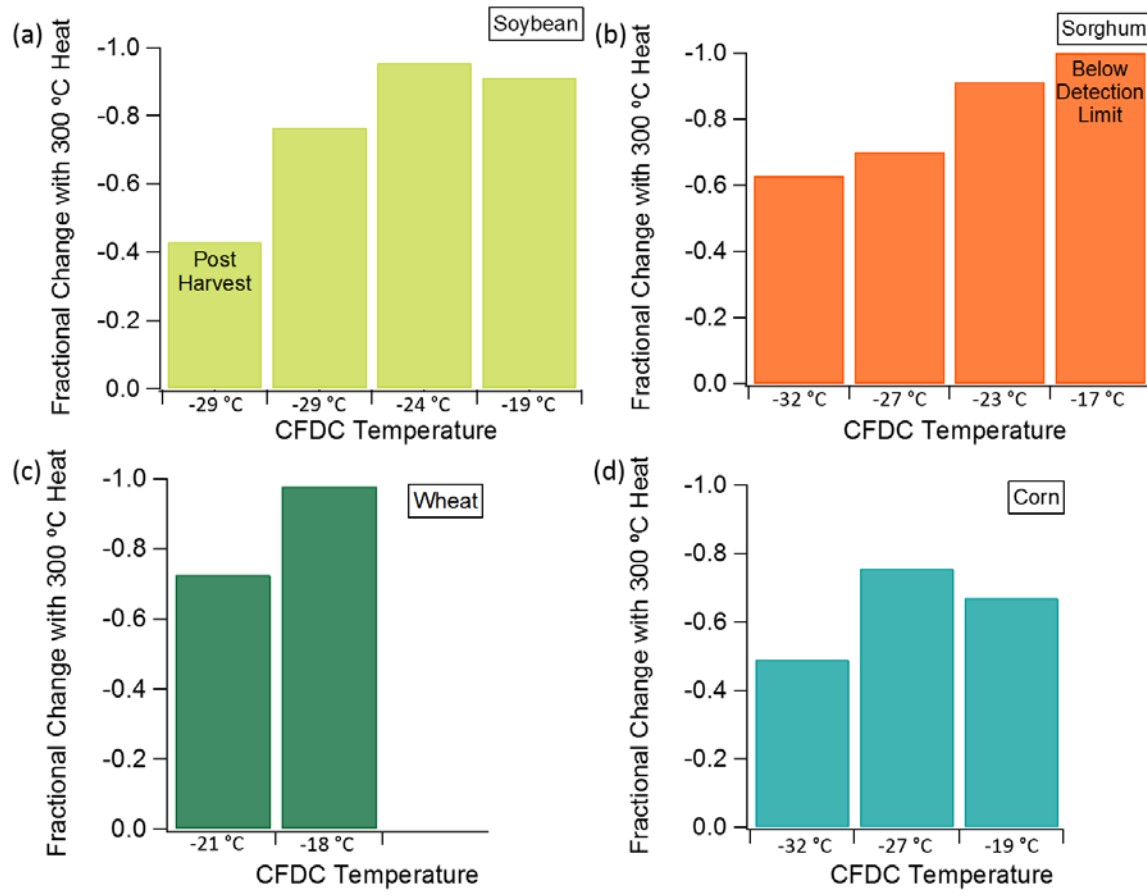

**Figure 6. The fractional change in INP number concentrations due to heating at 300 °C for four crop harvests including soybean (a), sorghum (b), wheat (c), and corn (d). The fractional change is shown for each CFDC operating temperature (x-axis) where measurements were made. Heating was done in situ using a heating tube upstream of the CFDC.**


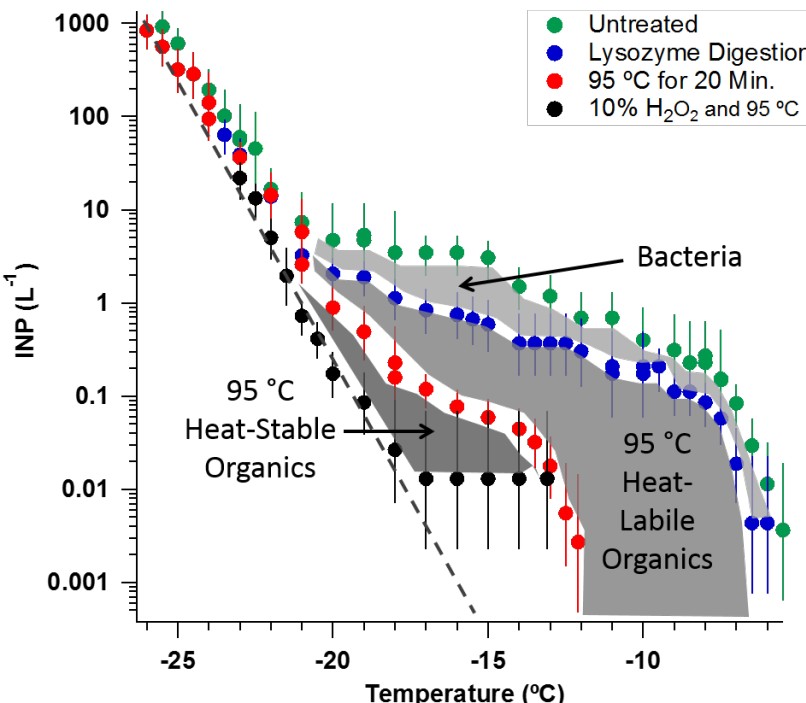

**Figure 7. INP number concentrations resulting from a wheat harvest on June 30, 2015 as measured by the IS (green), treating the wash water with lysozyme to selectively remove bacterial INP (blue), after heating to 95 °C for 20 minutes (red), and after peroxide digestion and heating to 95 °C (black). The reduction in INP concentrations by removal of bacteria, heat-labile, and heat-stable organics can be seen by the shaded areas. The dashed black line is representative of the likely underlying mineral INP spectrum.**





**Figure 8.** Measured CFDC (top row), IS (middle row), and IS behind a 2.5 µm cyclone (bottom row) INP number concentrations plotted against predicted INP number concentrations using the D10 (left), D15 (middle), and T13 (right) parameterizations. The markers are colored by the different harvests, the size of the CFDC square markers indicates if the concentrator was used (smaller squares) or not (larger squares). The grey dashed line represents a 1:1 line for measured versus predicted INP.