# Peer review of "Agricultural harvesting emissions of ice nucleating particles"

_Atmospheric Chemistry and Physics, 2018_

## Referee Comment (RC1) · Anonymous Referee #1 · 28 May 2018

The manuscript presents an investigation of airborne ice nucleating particles (INP) close to harvest operations for different crops at two sites in Kansas and Wyoming. The authors employed a variety of instruments, including modifications of them, to shed light on the characteristics of INP likely aerosolised by the operations (e.g. fluorescence, heat resistance, elemental composition, etc.). The lack of a consistently employed experimental protocol limits the scope for direct comparison between crops and makes for a slightly confusing reading experience. In this sense, I appreciated the 'Conclusions' section as a helpful summary of the most important findings.

My only major concern regarding the science is the lack of control measurements before the harvesting operations of sorghum, wheat and corn. The current interpretation of the data implicitly assumes that INP observed during harvest operations were

aerosolised by the operations. However, Figure 2 suggests a relevant contribution of INP and fluorescent particles were already present before the soybean harvest. Soybeans were harvested in the middle of October 2014, one day before sorghum. Hence the background of INP before sorghum harvest might have been similar to the pre-soybean harvest. However, wheat was harvested end of June and beginning of July in 2015, and corn in September 2015 at a different location from all the other crops. Harvest operations "usually lasted 2-4 hours" (page 2, lines 31-32). I assume the instruments had been installed before that. Why were no pre-harvest and/or post harvest measurements done, at least with the CFDC? As I understand, its lower limit of detection with the concentrator in place is close to 0.002 INP/L for an integrated sampling period of 10 min.

minor issues

page 8, line 1: perhaps change "be contributing" to "have contributed"

page 12, line 26: change "rust-infected the wheat" to "the rust-infected wheat"

Figure S2: The black lines connecting rain hat, aerosol concentrator, and CFDC would benefit from arrow ends indicating flow direction. Also, why are they entering the CFDC at different points and are joint inside it? Did they not connect to a common inlet of the CFDC? Perhaps add to the drawing also the ice particle collector at the outlet side of the CFDC.

―――――――――――――――――――――

---

## Referee Comment (RC2) · Anonymous Referee #2 · 15 Jun 2018

In the submitted work Suski et al. present ice nucleating particle (INP) measurement data from emissions associate with harvesting wheat, soybean, and sorghum crops at two locations in the plains east of the Rocky Mountains. I find the manuscript well written and clear. The subject is certainly very interesting, especially given the intense agricultural activities in this region of the United States. I would recommend this manuscript for publication subject to the authors addressing a few minor points.

Furthermore, I suggest a few points that may be considered if future measurements of a similar nature are undertaken. Although, I understand the challenges of rigidly constrained ambient measurements I do agree with Anonymous Referee # 1 that the link to a clean ambient, "control" context is not clear cut. Similarly, the measurements raise a lot of open questions about links to landscape and agricultural evolution and

larger synoptic scales and beyond.

Minor Comments:

• It is a bit strange that the first figures referred to in the manuscript are actually in the supplement. Perhaps an initial figure that summarized a bit of the experimental parameters could/should be included? Perhaps a map with the sites located and/or a photo that would give the reader an impression of the landscape and/or emission plumes?

Notes on figures S1, S2:

S1: Could this plot be turned into a box plot with the added axes indicating distance, in addition to the lat/long currently indicated?

S2: It would be useful to also indicate the height above ground level of the collection (rain hat and concentrator) inlets? These numbers could also be added to the text.

• Page 3, line 9; I suggest that the wording be changed to, "Ice nucleating particle concentrations were measured *online* with the ....." to distinguish the CFDC measurements from the IS.

• Page 3, line 11; I suggest that, "coated with ice at different temperatures" be changed to, " coated with ice *and held* at different temperatures". I believe the ice coating is done with the walls at a single temperature.

• The use of the 2.4 $\mu$m impactor and 2.5 $\mu$m cyclone is mentioned with respect to the two INP measurement techniques. However, given the nature of the emissions one would expect a sizable number of large particles. The authors do not show any particle size distributions (perhaps would be a useful addition to the supplement) or to my reading comment about how many particles might be left unobserved given the size cutoffs. While the very largest particles likely sediment out quickly and thus may not readily affect clouds, what about particles closer to the cutoffs? A short discussion of this would benefit the manuscript.

• A justification of the concentration factor (CF) of $90 \pm 3$ that is used for all measurements is lacking. The authors argue that CF calculations are difficult but do not present a clear explanation of why the choice that is made is considered representative for all measurements. An additional sentence is needed.

• Page 5, lines 5-6: The parenthetical statement, "to ensure maintenance of activity of K-feldspar, if present" does not obviously follow from an addition of 2 mM KCl. Perhaps I am ignorant, but I would suggest a sentence of explanation would add clarity.

• For the chemical composition analysis using the SEM-EDX a 2.9 $\mu$m impactor is used after the CFDC. To my understanding that leaves a range of particles between 2.4 $\mu$m and 2.9 $\mu$m that may include INP that will not be collected for analysis. Can the authors comment on the impact of this gap? Is it likely to in anyway impact results? It might be helpful if they state the median or mean size to which the ice crystals grow when measured leaving the OPC. I guess once out of the controlled chamber the ice crystals will be evaporating and shrinking quickly, how small will they get before arriving at the impactor?

• Throughout the manuscript $n_{0.5\mu m}$ is used, expect in Figure 5 when n500 is used. The figure labels should be modified for consistency.

• A general comment on figures: I find the size distinction between No Concentrator and Concentrator data points difficult to distinguish. Although, likely the authors have tried many combinations I wonder if the difference could be slightly amplified?

• Figure 1: The light green of (a) makes the symbols more difficult to distinguish than the other color choices.

• In Figures 3 and 4: It would be helpful to state the number of particles that were analyzed to arrive at the pie charts. Are these numbers arrived at using a particle by particle analysis, or does the method allow some type of averaging over the entire sample? Please also address these points in the text.

• Figure 5: See n500 comment above.

General comments to be addressed now or in future work:

The current manuscript raises many open questions in my mind, some of which may be addressed, but likely many which cannot be given the current data set. I briefly summarize some of these, perhaps some of which the author's could address or would be interested in incorporating as points of discussion. The most obvious question, is how do the measurements summarized here compare to conditions which would be found given other landscape contexts. Specifically, are there other measurements in the literature which would suggest how these observations might compare to a native landscape, which I assume was prairie? How about to other forms of agriculture? For example, this part of the US has seen a large homogenization of crops over the last half-century, how does that affect such emissions. Also, it is unclear exactly what kind of crops are being grown and harvesting is being done. For example are the crops "roundup ready", are the fields heavily treated with pesticides during the growing season? Also is the harvesting being done for silage or is just the main fruit/seed of the plant being harvested? How much residual vegetation is being left behind after the harvesting? For example are the corn fields cut completely to the base of the stalks or are only the ears of corn harvested and the stalks discarded onto the ground? These types of variables will significantly influence the amount of vegetation undergrowth, residual vegetation, and access to bare soil surface. One could envision an entire slate of field measurements to dig into some of these types of questions.

Finally, it remains unclear to me how relevant the ground based measurements are for cloud level processes. What is the anticipated spatial and temporal scale at which these emissions will remain relevant. Clear answers are beyond the scope of this manuscript, but perhaps there are existing modeling studies to which the authors could point.
* * *
[Figure]

2018.

---

## Author Comment (AC1) · 21 Aug 2018

We would like to thank the referees for their comments and suggestions. Our responses follow the referee comments.

Response to Anonymous Referee #1

My only major concern regarding the science is the lack of control measurements before the harvesting operations of sorghum, wheat and corn. The current interpretation of the data implicitly assumes that INP observed during harvest operations were aerosolised by the operations. However, Figure 2 suggests a relevant contribution of INP and fluorescent particles were already present before the soybean harvest. Soybeans were harvested in the middle of October 2014, one day before sorghum. Hence

the background of INP before sorghum harvest might have been similar to the presoybean harvest. However, wheat was harvested end of June and beginning of July in 2015, and corn in September 2015 at a different location from all the other crops. Harvest operations "usually lasted 2-4 hours" (page 2, lines 31-32). I assume the instruments had been installed before that. Why were no pre-harvest and/or post harvest measurements done, at least with the CFDC? As I understand, its lower limit of detection with the concentrator in place is close to 0.002 INP/L for an integrated sampling period of 10 min.

"The current interpretation of the data implicitly assumes that INP observed during harvest operations were aerosolized by the operations" We appreciate the Reviewer's point, but such an interpretation was not our intention. The only thing we are saying is these are the concentrations of INPs during harvests in agricultural regions. These include direct emissions from the targeted harvest as well as background INP from regional harvesting. There is no way to get "clean" measurements during harvesting because even if the farm we are at isn't harvesting, neighboring farms are harvesting. That's why our background measurement is described in the text as a regional harvesting background and not as a "clean" pre-harvesting sample.

Additionally, it was not possible to get this measurement for all of the harvests because we were in a mobile lab run on generators. When the farmers decided to harvest a crop, (which is done minutes to an hour before harvesting) we had to drive the truck to the field to be harvested, set-up, and get the CFDC instrument iced and ready. This takes about an hour. Therefore, we did not have time to get background measurements pre-harvest for every crop. Further, after harvesting was finished it was clear from the CFDC measures and aerosol counts that the harvested fields remained a significant source of INPs due to the lofting of pulverized plant tissues and soil dust. It could also be seen readily (Fig. 1 and 2 Attached).

Fig. 1.: "Dust" comprised of pulverized wheat creating an artificial sunset and lingering on the site post the Wheat1 harvest. Fig. 2: Plant tissue fragments lit by a flashlight
beam and lofted by strong winds in the sorghum field post-harvest.

During the corn harvest we were able to run a filter upwind of the field (extra generator). However, when another harvester could be seen operating in an upwind depression from that position. Because of this we didn't include the results from this filter, but in light of the Reviewer's concerns we will add it (see Table 2); it shows relatively low INP concentrations compared with those found immediately downwind of harvesting.

These measurements are intended to give an upper bound of INP emitted from harvests with the assumption that there are other INP sources present in the "background", but the bulk of the INP signal is directly emitted from harvesting based on the sheer number of particles emitted versus background INP, which are generally very low especially at modestly supercooled temperatures.

minor issues page 8, line 1: perhaps change "be contributing" to "have contributed" This has been changed.

page 12, line 26: change "rust-infected the wheat" to "the rust-infected wheat" This has been changed.

Figure S2: The black lines connecting rain hat, aerosol concentrator, and CFDC would benefit from arrow ends indicating flow direction. Also, why are they entering the CFDC at different points and are joint inside it? Did they not connect to a common inlet of the CFDC? Perhaps add to the drawing also the ice particle collector at the outlet side of the CFDC.

We have added arrows to indicate the flow direction.

Yes, the 3 different sampling lines are all connected to one inlet on the CFDC. The figure has been modified to make that more clear.

The impactor was added to the drawing.

Response to Anonymous Referee #2

ACPD
Although, I understand the challenges of rigidly constrained ambient measurements I do agree with Anonymous Referee # 1 that the link to a clean ambient, "control" context is not clear cut. Similarly, the measurements raise a lot of open questions about links to landscape and agricultural evolution and larger synoptic scales and beyond. Please see our response to this point above, and, yes, we agree wholeheartedly that the measures represent an integration of many inputs/sources, from the immediate, to local and regional agricultural ecotypes, to synoptic. But our measures do, undeniably, provide the first measures of INP spectra (upper bounds, as mentioned above) downwind of harvesting of several major crops, and not only in the U.S.

Minor Comments: It is a bit strange that the first figures referred to in the manuscript are actually in the supplement. Perhaps an initial figure that summarized a bit of the experimental parameters could/should be included? Perhaps a map with the sites located and/or a photo that would give the reader an impression of the landscape and/or emission plumes? These figures are included in the supplemental because they are not scientifically necessary figures, but add to the understanding of the experimental set-up. We agree with Rev 2's suggestion that pictures will be valuable to help to set the scene, and we do have a good selection showing all harvests, the plumes produced, and the fields before and afterward. Since there are a number of them, we will place them in the supplement as well.

Notes on figures S1, S2: S1: Could this plot be turned into a box plot with the added axes indicating distance, in addition to the lat/long currently indicated? Do you mean the distance from the combine to the mobile lab? This GPS trace is included just to illustrate how the sampling was done and to show that we saw pulses of particles based on wind direction and the location of the combine.

S2: It would be useful to also indicate the height above ground level of the collection (rain hat and concentrator) inlets? These numbers could also be added to the text. This information has been added to the text and figure.
Page 3, line 9; I suggest that the wording be changed to, "Ice nucleating particle concentrations were measured online with the ....." to distinguish the CFDC measurements from the IS. This has been changed.

Page 3, line 11; I suggest that, "coated with ice at different temperatures" be changed to, "coated with ice and held at different temperatures". I believe the ice coating is done with the walls at a single temperature. This has been changed for clarity.

The use of the 2.4  $\mu$ m impactor and 2.5  $\mu$ m cyclone is mentioned with respect to the two INP measurement techniques. However, given the nature of the emissions one would expect a sizable number of large particles. The authors do not show any particle size distributions (perhaps would be a useful addition to the supplement) or to my reading comment about how many particles might be left unobserved given the size cutoffs. While the very largest particles likely sediment out guickly and thus may not readily affect clouds, what about particles closer to the cutoffs? A short discussion of this would benefit the manuscript. We do address this when we look at the IS measurements with and without the impactor. This data is supplied in Table 2. You can see that there is significantly less INPs at certain temperatures when using a cyclone or impactor; however, as you mention these larger particles are likely not making it to cloud level. We have added some discussion of this. It now reads "The corn and wheat IS data were sampled through a 2.5  $\mu$ m cyclone, in addition to open-faced Nalgene sterile filter units, to limit the size range of particles that were collected. While the use of the cyclone will not capture the IN activity of larger particles, these larger particles will sediment out faster than smaller particles and likely do not make it to cloud level. Therefore, the use of the cyclone and impactor offer a better representation of particles that could impact clouds. IS data with and without the cyclone is provided in Table 2 for a comparison of INP concentrations with and without larger particles."

A justification of the concentration factor (CF) of 90  $\pm$  3 that is used for all measurements is lacking. The authors argue that CF calculations are difficult but do not present a clear explanation of why the choice that is made is considered representative for

**ACPD**
all measurements. An additional sentence is needed. Especially considering that the sampling period was generally one during harvesting, and that the CF factor determined when aerosol concentrations were stable is very close to the value measured and applied previously by Tobo et al. (2013), we feel confident that we are justified in applying the determined value. We have measured the CF factor in some other locations, duplicating a range from 90 to 110 in many circumstances, but do not wish to report those additional results here. We have added the following sentences and modified the section describing the CF on page 4 as follows: "It was important to use a time period within the overall harvesting experimental period that included stable aerosol concentrations during periods on and off the concentrator. Therefore, the CF calculated during a pre-soybean harvest period in Colby, KS (CF = 90  $\pm$  3) was used as the CF for all of the harvests. We may note that this value is within 15% of the value found by Tobo et al. (2013) in prior studies with the same aerosol concentrator and CFDC instrument. This is physically expected if most INPs reside in the size range above 0.5 mm."

Tobo, Y., Prenni, A. J., DeMott, P. J., Huffman, J. A., McCluskey, C. S., Tian, G., Poehlker, C., Poeschl, U., and Kreidenweis, S. M.: Biological aerosol particles as a key determinant of ice nuclei populations in a forest ecosystem, Journal of Geophysical Research-Atmospheres, 118, 10100-10110, 10.1002/jgrd.50801, 2013.

Page 5, lines 5-6: The parenthetical statement, "to ensure maintenance of activity of K-feldspar, if present" does not obviously follow from an addition of 2 mM KCI. Perhaps I am ignorant, but I would suggest a sentence of explanation would add clarity. We have added this to the text. "Prior tests on dilute suspensions of pure K-feldspar found that use of deionized water reduced IN activity, presumably due to desorption of K+ ions; use of a suspension containing  $\geq$ 0.1 mM K+ prevented this and so was used for dilutions."

For the chemical composition analysis using the SEM-EDX a 2.9  $\mu m$  impactor is used after the CFDC. To my understanding that leaves a range of particles between 2.4  $\mu m$

**ACPD**
and 2.9  $\mu$ m that may include INP that will not be collected for analysis. Can the authors comment on the impact of this gap? Is it likely to in anyway impact results? It might be helpful if they state the median or mean size to which the ice crystals grow when measured leaving the OPC. I guess once out of the controlled chamber the ice crystals will be evaporating and shrinking quickly, how small will they get before arriving at the impactor? The minimum size of ice crystals and the maximum size of aerosol particles being sampled are the critical parameters for effectively capturing activated ice crystals while limiting any overlap with larger aerosols. The gap between the upstream impactor cut-size and the ice crystal impactor is therefore vital, especially considering the fact that cut-sizes represent 50% points and are imperfect. We are much less concerned that any INPs are missed that lie between the cutpoints. For the conditions represented in this paper, ice crystals are always expected to be above a size of about 3 um (DeMott et al., 2015). This is largely driven by the fact that liquid droplets are achieving this size prior to entering the instrument evaporation region, so that freezing is occurring already at such sizes, and ice crystal growth is ensuing from that point until detection. Most ice crystals are detected in the uppermost bins of the optical particle counter, representing sizes 5 um and larger (larger particles are placed into the last bin). The reviewer is correct that we have no assurances that these ice crystal sizes are maintained to the impactor, but these are the sizes measured just a few cm distant from the impaction grid, so we expect high collection efficiencies. That the technique works has been demonstrated in numerous prior papers.

DeMott, P. J., Prenni, A. J., McMeeking, G. R., Sullivan, R. C., Petters, M. D., Tobo, Y., Niemand, M., Moehler, O., Snider, J. R., Wang, Z., and Kreidenweis, S. M.: Integrating laboratory and field data to quantify the immersion freezing ice nucleation activity of mineral dust particles, Atmospheric Chemistry and Physics, 15, 393-409, 10.5194/acp-15-393-2015, 2015.

Throughout the manuscript  $n0.5\mu$ m is used, expect in Figure 5 when n500 is used. The figure labels should be modified for consistency. The labels have been modified.
A general comment on figures: I find the size distinction between No Concentrator and Concentrator data points difficult to distinguish. Although, likely the authors have tried many combinations I wonder if the difference could be slightly amplified? If the small symbols are made smaller, they are hard to see and if the larger symbols are made larger they cover each other. We have made them as distinct as possible, while still being able to see everything.

Figure 1: The light green of (a) makes the symbols more difficult to distinguish than the other color choices. This color scheme is consistent throughout the paper and we would like to keep the color.

In Figures 3 and 4: It would be helpful to state the number of particles that were analyzed to arrive at the pie charts. Are these numbers arrived at using a particle by particle analysis, or does the method allow some type of averaging over the entire sample? Please also address these points in the text. The number of analyzed particles has been added to the figure captions and the text has been modified to read "Analysis was done by analyzing individual particles on the filters (73 particles for the sorghum sample and 67and 72 particles for the corn samples). Characteristic combinations of elements were identified and then used to group the individual particles into classes."

Figure 5: See n500 comment above. It has been changed.

General comments to be addressed now or in future work: The current manuscript raises many open questions in my mind, some of which may be addressed, but likely many which cannot be given the current data set. I briefly summarize some of these, perhaps some of which the author's could address or would be interested in incorporating as points of discussion. The most obvious question, is how do the measurements summarized here compare to conditions which would be found given other landscape contexts. Specifically, are there other measurements in the literature which would suggest how these observations might compare to a native landscape, which I assume was
prairie? This is a very pertinent comment, and we are addressing this point in an upcoming publication. The forthcoming work will, in part, compare the harvest sampling as well as measurements in various natural landscapes, including native grasslands, as mentioned. But in short, the harvest emissions have much higher concentrations than natural landscapes.

How about to other forms of agriculture? For example, this part of the US has seen a large homogenization of crops over the last half-century, how does that affect such emissions. Good question. The increasing homogenization of crops grown in this part of the U.S. may not have changed the overall amount of INPs released compared with the greater heterogeneity of species and strains grown previously. This is because previous crops would have produced a mix of both higher and lower emissions. For example, Georgakapoulos and Sands (1992) recorded a 5,000-fold range in populations of IN P. syringae among 23 barley lines and cultivars grown in Bozeman, Montana. However, greater patchiness of the landscape would have required a longer period over which harvesting emissions occurred in each region due to differences in maturation times. This discussion has been added to the conclusions section.

Also, it is unclear exactly what kind of crops are being grown and harvesting is being done. For example are the crops "roundup ready", are the fields heavily treated with pesticides during the growing season? Also is the harvesting being done for silage or is just the main fruit/seed of the plant being harvested? The fields harvested at the Colby and SAREC research stations were planted with a range of cultivars, since these are experimental stations. The corn was Roundup ready, but we do not know the status of the other crops. All were harvested for their grain. Unfortunately, we cannot compare the differences in these practices with our limited dataset.

How much residual vegetation is being left behind after the harvesting? For example are the corn fields cut completely to the base of the stalks or are only the ears of corn harvested and the stalks discarded onto the ground? These types of variables will significantly influence the amount of vegetation undergrowth, residual vegetation, and
access to bare soil surface. One could envision an entire slate of field measurements to dig into some of these types of questions. The amount of plant left behind is crop specific as is the amount of soil that is kicked up during harvesting. Corn harvesting for instance barely disturbed the soil as just the head of the corn was removed; however, the soybeans were much lower to the ground and their harvesting resulted in more surface soil being disturbed. This crop variability was taken into account and is briefly mentioned in the text. We sampled a variety of crops to capture some of this variability, but it is by no means a complete characterization of how farming methods impact emissions. But, again, Reviewer 2 has raised valuable questions that other readers will, no doubt, also consider. To provide some clarity and a visual aid for readers, we have, therefore added images of each harvest, that show before, during and after effects of the operations, to the supplementary information.

Finally, it remains unclear to me how relevant the ground based measurements are for cloud level processes. What is the anticipated spatial and temporal scale at which these emissions will remain relevant. Clear answers are beyond the scope of this manuscript, but perhaps there are existing modeling studies to which the authors could point. To our knowledge, there are not any modeling studies that look specifically at harvest emissions and their impacts. Most work just looks at the mass of emitted material. This work was meant as a means to characterize emissions at the emission source and future studies should look at airborne measurements to learn more about the spatial and temporal evolution of these emissions.

A couple of corrections that were detected in the course of responding to the reviewers:

Section 3.1. line 27. We had written "In this study, average number concentrations of 0.3 L-1 and 3.6 L-1 were measured with the IS and CFDC, respectively, at this temperature". In fact the correct value for comparison with the BioSampler used in Garcia et al. was from the open-faced filter, which was 8.1 L-1. We have now used this and removed the sentence discussing the reduction caused by use of the cyclone.
Table 2. The value for INPs active at -15 C during the wheat harvest was incorrectly entered. It was given as 0.02 but was, in fact, 0.22. This has been changed.
**Fig. 1.** "Dust" comprised of pulverized wheat creating an artificial sunset and lingering on the site post the Wheat 1 harvest.